# The Relationship between Angiogenic Factors and Energy Metabolism in Preeclampsia

**DOI:** 10.3390/nu14102172

**Published:** 2022-05-23

**Authors:** Alejandra Abascal-Saiz, Marta Duque-Alcorta, Victoria Fioravantti, Eugenia Antolín, Eva Fuente-Luelmo, María Haro, María P. Ramos-Álvarez, Germán Perdomo, José L. Bartha

**Affiliations:** 1Department of Obstetrics and Gynecology, Division of Maternal and Fetal Medicine, La Paz University Hospital, Paseo de la Castellana 261, 28046 Madrid, Spain; alejandra_as@hotmail.com (A.A.-S.); eantolin@salud.madrid.org (E.A.); 2Department of Clinical Chemistry, La Paz University Hospital, 28046 Madrid, Spain; marta.duqueal@salud.madrid.org; 3Department of Pediatric Hematology and Oncology, Hospital Infantil Universitario Niño Jesus, 28009 Madrid, Spain; v.fioravantti@gmail.com; 4Department of Biochemistry and Molecular Biology, Faculty of Pharmacy, CEU-San Pablo University, 28668 Madrid, Spain; eva.defuenteluelmo@ceu.es (E.F.-L.); maria.harogarcia@ceu.es (M.H.); pramos@ceu.es (M.P.R.-Á.); 5Unidad de Excelencia Instituto de Biología y Genética Molecular, University of Valladolid-CSIC, 47003 Valladolid, Spain; perdomogm@yahoo.com

**Keywords:** angiogenic markers, gestational hypertension, correlation, lipid metabolism, esterification, oxidation, triglyceride, carbohydrate metabolism, placental villous explants, placental function

## Abstract

Antiangiogenic factors are currently used for the prediction of preeclampsia. The present study aimed to evaluate the relationship between antiangiogenic factors and lipid and carbohydrate metabolism in maternal plasma and placenta. We analyzed 56 pregnant women, 30 healthy and 26 with preeclampsia (including early and late onset). We compared antiangiogenic factors soluble Fms-like Tyrosine Kinase-1 (sfLt-1), placental growth factor (PlGF), and soluble endoglin (sEng)), lipid and carbohydrate metabolism in maternal plasma, and lipid metabolism in the placenta from assays of fatty acid oxidation, fatty acid esterification, and triglyceride levels in all groups. Antiangiogenic factors sFlt-1, sFlt-1/PlGF ratio, and sEng showed a positive correlation with triglyceride, free fatty acid, and C-peptide maternal serum levels. However, there was no relationship between angiogenic factors and placental lipid metabolism parameters. Free fatty acids were predictive of elevated sFlt-1 and sEng, while C-peptide was predictive of an elevated sFlt1/PlGF ratio. The findings in this study generate a model to predict elevated antiangiogenic factor values and the relationship between them with different products of lipid and carbohydrate metabolism in maternal serum and placenta in preeclampsia.

## 1. Introduction

### 1.1. Etiopathogenesis of Preeclampsia: Current State

Preeclampsia is defined by a new onset of gestational hypertension at or after 20 weeks’ gestation accompanied by one or more of the following new-onset conditions: proteinuria, any maternal organ dysfunction (renal, hepatic, neurological or hematological) or uteroplacental dysfunction (such as fetal growth restriction, abnormal umbilical artery Doppler wave form analysis, or stillbirth) [1]. Preeclampsia continues to be one of the major causes of maternal morbidity and mortality worldwide, with a prevalence of around 5% [2,3] and around 50,000 maternal deaths worldwide per year [4]. Although risk factors are well established, etiopathogenesis remains uncertain. The current theory postulates a two-stage mechanism [5]. The first occurs in early gestation, with a deficient invasion of trophoblastic cells causing an inadequate remodeling of placental spiral arteries with the subsequent placental hypoperfusion and lower uteroplacental flow. The second stage involves the maternal response with increased oxidative stress, syncytial necrosis, and endothelial dysfunction causing the release of apoptotic cells and syncytiotrophoblast microparticle debris to maternal serum. This condition results in an imbalance between circulating anti/angiogenic factors, favoring the inhibition of angiogenesis during placental endothelial hypoxia. All of these changes result in a stimulating inflammatory response by increasing tumor necrosis factor-α (TNF-α) and interleukin-6 (IL-6), leading to an abnormal placental vascularization (and in some cases to a placental insufficiency with intrauterine growth restriction), multiorganic endothelial dysfunction, and development of clinical features [2,6,7,8].

However, this theory cannot explain the pathogenesis of late-onset preeclampsia (LOPE) (those that debut after the 34th week of pregnancy), in which there seems to be a relative placental insufficiency with a discordance between normal maternal perfusion and the metabolic requirements of the placenta and fetus. In recent years, multi-step models have emerged, trying to solve these unknowns. Some postulate that an impaired trophoblast immune tolerance or pregravid maternal conditions such as obesity, hypertriglyceridemia, insulin resistance, diabetes, or autoimmune disorders could predispose to a chronic vascular inflammation, leading to unremodeled spiral arteries and poor placentation [3,9,10,11,12]. Other theories to explain those cases of LOPE associated with normal fetal growth propose that the increased placental mass outgrows uterine capacity, producing compression and congestion of the villi with subsequent malperfusion. Syncytiotrophoblast stress in late pregnancy added to an increased susceptibility to chronic vascular inflammation is another theory [13,14]. Nevertheless, even if early-onset preeclampsia (EOPE, those that debut before the 34th week of pregnancy) and LOPE have different pathogenesis, they would share a common final metabolic pathway leading to placental dysfunction and ischemia that increase the antiangiogenic factors in a final stage.

### 1.2. Antiangiogenic Factors as Biomarkers in Preeclampsia

In basal conditions, vascular endothelial growth factor (VEGF) is essential in all stages of placental angiogenesis, promoting placental endothelial proliferation, migration, and tube formation through the activation of a complex signaling network involving the MAPK, the PI3K/Akt1, and the endothelial nitric oxide synthase (eNOS) pathways [15,16]. Placental growth factor (PlGF) has a synergistic angiogenic effect interacting with the VEGF receptor (VEGFR-1), therefore decreasing from the first trimester in preeclampsia [2]. The involvement of antiangiogenic factors expressed in the syncytiotrophoblast, such as soluble fms-like tyrosine kinase-1 (sFlt-1) or soluble endoglin (sEng), with the development of preeclampsia is also well known. The quantification of these biomarkers allows a differential diagnosis between preeclampsia and other hypertensive pregnancy disorders. Furthermore, they predict its appearance, severity, and the probability of adverse outcomes [17,18]. They operate blocking angiogenesis through different mechanisms which combined lead to endothelial dysfunction. sFlt-1 (the soluble form of VEGFR-1) binds to VEGF and PlGF, avoiding their interaction with their membrane receptors. sFlt-1 is detectable up to five weeks before the onset of preeclampsia symptoms, but its levels during pregnancy fluctuate, making it difficult to interpret [19]. On its part, sEng (the soluble form of a co-receptor for transforming growth factor β1, TGF-β1) binds to circulating TGF-β1, decreasing its availability to interact with its membrane receptors [20]. An increase in sEng has been detected even two or three months before the development of preeclampsia, being closely related to secondary symptoms such as seizures, liver dysfunction, and hypercoagulation [7,19,21]. Measurement of serum sEng, sFlt-1, and PlGF improves the sensitivity and specificity of prediction of preeclampsia and its severity [21].

### 1.3. Purpose of the Study

It is not yet well established if these antiangiogenic factors are simple biomarkers of placental dysfunction in early pregnancy or whether the local decidual dysregulation of such antiangiogenic markers mediates early defective placentation as the first stage of preeclampsia [13]. The mechanisms of regulation of these antiangiogenic factors remain unexplored, as do their relationship with maternal and placental lipid metabolism. It is well established that in preeclampsia associated with pregravid obesity or diabetes mellitus, antiangiogenic factors have a worse profile compared to those that only have preeclampsia [14], thereby making it important to clarify a potential relationship with maternal or placental metabolic markers that could act as risk cofactors in the development of preeclampsia. Based on the fact that preeclamptic women exhibit changes in lipid and carbohydrate metabolism, both at the serum and placental level, our study aims to evaluate the relationship of serum antiangiogenic factors with energy metabolism parameters in maternal plasma and placenta, comparing both healthy pregnant women and those with preeclampsia. This is of unique relevance in LOPE given its association with pre-existing metabolic syndrome but also due to a reduced clinical value of antiangiogenic factors in LOPE compared to EOPE [22]. Another aim of the study is to compare these parameters in EOPE vs. LOPE. Additionally, we seek to elucidate if these metabolic parameters could add predictive value to the already known antiangiogenic factors.

## 2. Materials and Methods

### 2.1. Study Participants

A prospective observational study was carried out at La Paz University Hospital (Department of Obstetrics and Gynecology) for two years. This study was approved by the Local Ethical Committee, and all the participants signed informed consent forms.

Fifty-six women were included and allocated into three groups, 30 healthy patients as a control group and 26 with preeclampsia (11 EOPE and 15 LOPE). Ten out of fifty-six patients were twin pregnancies, six of which developed preeclampsia. To avoid the potential effects of labor contractions on placental metabolism, all included patients had deliveries performed by cesarean section due to clinical reasons not affecting placental metabolism or perfusion, as shown in Table 1.

The inclusion criteria for healthy control subjects were as follows: normal blood pressure during pregnancy, pregnancy at term, no medical history of chronic metabolic diseases or any pathology that could involve lipid or carbohydrate metabolism disorders, and no complications during pregnancy. The diagnosis and classification of preeclampsia were made with the criteria provided by the International Society for the Study of Hypertension in Pregnancy (ISSHP) [1]. Specific exclusion criteria included major fetal anomalies and women with a history of long-chain 3-hydroxyacyl-coA dehydrogenase (LCHAD) deficiency, mitochondrial trifunctional protein (TFP) deficiency, or acute fatty liver of pregnancy.

### 2.2. Sample Collection

Fasting blood samples were extracted previously to the cesarean section on the day of delivery. Anti/angiogenic factors as soluble sfLt-1 and PlGF were determined by automated methods by electrochemiluminescence using the *Elecsys sFlt-1* and *Elecsys PlGF* assays (Roche Diagnostics Kit, Mannheim, Germany) to calculate the sfLt-1/PlGF ratio as a preeclampsia predictor. The *Elecsys sFlt-1* assay has a lower limit of detection of 10 pg/mL (measuring range, 10–85,000 pg/mL). The *Elecsys PlGF* assay has a lower limit of detection of 3 pg/mL (measuring range of 3 to 10 000 pg/mL). Intra-assay coefficients of variation for the control samples were <4.3% for sFlt-1 and <4.1% for PlGF, whereas inter-assay coefficients of variation were <5.6% and <4.6% for sFlt-1 and PlGF, respectively. Serum levels of sEng were determined in duplicate by ELISA (R&D Systems, Minneapolis, MN, USA) following the manufacturer’s instructions. The intra- and inter-assay coefficients of variations were 4.7% and 5.2%, respectively. The sFlt-1/PlGF cut-off of <38 was able to rule out preeclampsia within one week. The sFlt-1/PlGF cut-off of 85 is able to detect EOPE and 110 to detect LOPE [23,24].

Atherogenic indices were calculated by using the values of lipid profile parameters on the following formulas: Triglyceride/HDL ratio, Atherogenic Index of Plasma (AIP) calculated as Log10 (triglyceride/HDL); Cardiac risk ratio (CRR, also called Castelli risk index I) calculated as total cholesterol/HDL; Castelli risk index II (CRI-II) calculated as LDL/HDL. The fresh placenta was transported on dry ice to the laboratory within 2 h of delivery. Six 100 mg cotyledon fragments (chorionic villous explants) were collected removing decidual tissue, calcium deposits, and large vessels from regions near the chorionic plate and the basal plate. In both groups, explants were taken from central, intermediate, and peripheral locations to the umbilical cord insertion.

### 2.3. Placental Lipid Analysis

Measurement of fatty acid oxidation (FAO) in placental explants was performed according to the method previously described by our group [25,26,27,28]. It was quantified as nanomoles of [3H]-palmitate per gram of tissue per hour (nmol/g/h). Placental fatty acid esterification (FAE) was performed after thawing the explants. They were washed and homogenized in 500 μL of cold phosphate-buffered saline (PBS), and an aliquot of 100 μL was used to extract the lipid content following the method previously published [29,30] and was also quantified as nanomoles of [3H]-palmitate per gram of tissue per hour (nmol/g/h). As previously described [31], to quantify placental triglyceride concentrations, previously thawed placental fragments of 20 mg were homogenized in 400 μL of HPLC-grade acetone. To eliminate bias on the composition of the placenta and accurately establish the placental lipid proportion, placental lipid content was calculated as milligrams of triglyceride per milligrams of total placental proteins (TG/Prot ratio). The total placental results of each assay were obtained by calculating the mean of all the cotyledon fragments analyzed from each patient.

### 2.4. Statistical Analysis

All data were analyzed with SPSS 20.0. The normal distribution of variables was assessed by the Kolmogorov–Smirnov test. Student’s *t*-test (data were expressed as mean ± standard deviation) was used for continuous normally distributed variables, as was Mann–Whitney U test (data were shown as a median and interquartile range) for continuous non-normally distributed variables. Spearman’s correlation coefficient (rs) and simple linear regression analysis (R2) were used to evaluate and quantify the association between variables. Differences were considered significant at a *p* value < 0.05.

## 3. Results

### 3.1. Maternal, Obstetric and Perinatal Outcomes

Maternal characteristics and obstetric-perinatal outcomes of control and preeclampsia groups are shown in Table 1 (the characteristics of the EOPE vs. LOPE groups are shown in Appendix A). The demographic characteristics of both groups do not show significant differences except for a higher rate of primigravid pregnancies in the preeclampsia group and different indications of cesarean section in both groups. The gestational age at study, neonatal and placental weights were lower in the preeclampsia group, which can be explained by prematurity associated with the early termination of pregnancy in some and that nearly half of the preeclampsia cases presented fetal growth restriction.

### 3.2. Serum and Placental Analysis

#### 3.2.1. Lipids and Carbohydrates in Maternal Plasma

As shown in Table 2, maternal lipid parameters in patients with preeclampsia show a trend toward a profile associated with higher cardiovascular risk without reaching significant differences. Statistically significant differences were found in the levels of free fatty acids (FFA), which were 43% higher in patients with preeclampsia compared to healthy pregnant women. These differences remain significant when comparing controls versus EOPE or LOPE separately (*p* = 0.031 and *p* = 0.019 respectively), without observing a difference when analyzing the two subsets of preeclampsia. Concerning the analysis of both preeclampsia subgroups, the lipid profile worsened significantly in LOPE compared with EOPE, with higher triglyceride and LDL levels, and lower HDL levels. Moreover, when comparing controls only with the LOPE group, triglyceride levels were 34% higher in LOPE (*p* = 0.031), as well as the TG/HDL ratio (*p* = 0.013) and AIP (*p* = 0.011). None of these differences were observed when analyzing controls against EOPE. Instead, when comparing healthy pregnant women only with EOPE, HDL was 33% higher in the latter (*p* = 0.002), and all the atherogenic indices explored were higher in the control group (CRR, *p* = 0.027; CRI-II, *p* = 0.023; TG/HDL, *p* = 0.033, AIP, *p* = 0.043).

The only difference detected in the glycemic profile was in C-peptide, which was 41% higher in preeclampsia when compared to controls (*p* = 0.021), with no differences between both subgroups of preeclampsia.

#### 3.2.2. Antiangiogenic Factors in Maternal Plasma

When comparing the preeclampsia vs. control group, sFlt-1 and sEng antiangiogenic marker levels were four times higher in preeclampsia, and the angiogenic factor PlGF was three times lower. Consequently, the sFlt-1/PlGF ratio in patients with preeclampsia was significantly elevated (×6) when compared with healthy pregnant women (Table 2). Within the preeclampsia subgroups, PlGF was reduced in EOPE samples, approximately at a half when compared to LOPE (*p* = 0.041), resulting in an sFlt-1/PlGF ratio five times higher in EOPE than in LOPE (*p* = 0.008), with no significant differences in sEng levels.

#### 3.2.3. Placental Lipid Metabolism

No relevant differences were observed when assessing FAO and FAE (Table 3), neither in the comparison of control vs. preeclampsia, per preeclampsia subset, or in the comparison of controls with the subgroups of preeclampsia separately. On the contrary, the triglyceride level was found to be twice as high in the placentas of EOPE cases compared to LOPE (*p* = 0.026), even when adjusted by the TG/Prot ratio (*p* = 0.045).

### 3.3. Correlation of Maternal and Placental Metabolism with Antiangiogenic Factors

In the study of lipid metabolism in maternal serum, we did not find a significant association between antiangiogenic factors and total cholesterol, LDL, or HDL maternal serum levels. However, a statistically significant moderate positive correlation was found between maternal blood triglycerides and sFlt-1 or sEng levels (rs = 0.383 and 0.405, respectively; Table 4). Likewise, a strong significant positive relationship was demonstrated between the FFA levels and the sFlt-1, sFlt-1/PlGF ratio and sEng levels (rs = 0.575, 0.509 and 0.522, respectively). Concerning maternal plasma carbohydrate metabolism, C-peptide had a statistically significant moderate positive correlation with sFlt-1, sFlt-1/PlGF ratio and sEng (rs = 0.348, 0.394 and 0.466, respectively). We could not prove a correlation of antiangiogenic factors with placental lipid metabolism.

The best model for prediction of sFlt-1 levels (R2 = 0.57, *p* = 2 ×10 ^− 7^) included two variables: FFA levels and the presence of preeclampsia (R2 = 0.52 and 0.34, *p* = 2 ×10 ^− 4^ and 0.01, respectively) (Figure 1a). In case of the sFlt-1/PlGF ratio, the best model for prediction (R2 = 0.41, *p* = 4 ×10 ^− 5^) involved two variables: C-peptide levels and the presence of preeclampsia (R2 = 0.30 and 0.49, *p* = 0.02 and 0.001, correspondingly) (Figure 1c). On the other hand, the best model for prediction of sEngl levels (R2 = 0.46, *p* = 1 ×10 ^− 4^) comprised only one variable: FFA levels (R2 = 0.69, *p* = 1 ×10 ^− 4^) (Figure 1b).

## 4. Discussion

### 4.1. Main Findings

The heterogeneity in the clinical presentation of preeclampsia, especially in severe cases, together with a still unclear etiopathogenesis, renders the existence of a single predictive marker for the entire disease spectrum unlikely. Some new predictive cofactors, including those derived from metabolomic or proteomic studies, could improve their sensitivity and specificity [32], being especially relevant in LOPE, in which the antiangiogenic factors have a lower prediction rate [22]. Previous experimental studies have linked antiangiogenic factors with other maternal serum parameters, such as vitamin D deficiency [33], increased IL-6 or C-reactive protein [34], and decreased leptin levels [35]. To improve the predictive diagnosis of antiangiogenic factors, combination models of serum markers have been proposed. An example includes VEGF, sEng, PlGF, and soluble epidermal growth factor receptor (sEGFR). Furthermore, independent variables within a risk score model that included serum creatinine, platelet count, and sEng could predict severe features [36].

Our research has shown a relationship between the antiangiogenic factors currently used as predictors of preeclampsia (sFlt-1, sFlt-1/PlGF ratio, and sEng) and different parameters of lipid and carbohydrate metabolism in maternal serum. These relationships have allowed us to discover energy products that predict the elevation of these antiangiogenic markers, such as FFA for sFlt-1 or sEng, and C-peptide for the sFlt-1/PlGF ratio.

### 4.2. Interpretation of Results

#### 4.2.1. Relationship between Insulin Resistance and Preeclampsia

Preeclampsia can be considered a low-grade systemic inflammatory condition characterized by peripheral insulin resistance, a prediabetes-like state, and hypertension, with potential secondary multi-organ damage due to endothelial dysfunction. On the other hand, insulin resistance is considered a risk factor for the development of hypertension, both in pregnant and non-pregnant women [37,38,39]. The specific mechanism by which endothelial damage leads to hypertension in diabetic or insulin-resistant patients is not yet established. Inflammation and oxidative stress are suspected to play an important role in endothelial dysfunction [19,40] due to the angiogenic and insulin-dependent pathways that may influence each other [41].

From mid-pregnancy, insulin resistance appears physiologically. Lower levels of hyperinsulinism than in those found in gestational diabetes are sufficient to increase the risk of hypertension [38,40,42] independently of body mass index (BMI) [37] and mid-pregnancy blood pressure [38]. Some studies have proposed hyperinsulinism as a marker for the follow-up of patients with a higher risk of developing gestational hypertension in the third trimester [37,43], although it seems insufficient on its own. In addition to cellular metabolic changes, insulin has independent effects on vascularization: it contributes to peripheral vasodilation by regulating calcium transport [44] and endothelial production of nitric oxide [45], it promotes renal sodium retention, and it increases noradrenaline levels [37,41]. This influence on the autonomic nervous system (ANS) is reflected in the relationship of insulin resistance with the worsening of some ANS markers (noradrenaline and changes in heart rate variability on Holter ECG) [46].

In our study, insulin resistance in patients with preeclampsia can be reflected in the increase in maternal plasma of fasting C-peptide. Insulin and C-peptide are secreted in equimolar amounts by pancreatic B-cells. However, C-peptide more accurately reflects the pancreatic cell capacity for hormonal production and hyperinsulinism, as it is subjected to fewer metabolic changes. C-peptide is part of the pro-insulin molecule; it has a longer half-life in plasma than insulin (30 vs. 5 min), it is not affected by hemolysis, and its degradation is exclusively renal while insulin suffers hepatic, renal, placental, pulmonary, and muscular elimination [37,40]. Therefore, an increase in C-peptide in blood (>2.0 ng/mL) would reflect a state of hyperinsulinism [38,40], which correlated better than insulin resistance with antiangiogenic factors in our study, although both concepts are closely related. In other studies, the relationship between elevated antiangiogenic factors and increased HOMA-IR in preeclampsia has been established; however, we found a better correlation between the elevation of antiangiogenic factors and C-peptide. In a recent meta-analysis, an increase in C-peptide was associated with an increase in cardiovascular mortality in the general population, independently of other comorbidities [47]. Inflammation and atherogenesis seem to be involved in endothelial damage in cardiovascular diseases [48] and are also probably related to insulin resistance [49]. The relationship between C-peptide and the development of gestational hypertension is poorly studied. Yogev et al. established that the measurement of C-peptide as a reflection of insulin sensitivity in pregnancy is adequate, being a predictor of preeclampsia independent of BMI and fasting glucose [40]; while Yasuhi et al. demonstrated a relationship between the development of hypertension in mid-pregnancy and the elevation of C-peptide, theorizing an exaggerated response of B-cells to glycemic stimulation reflecting a lower insulin sensitivity [38]. On the other hand, other authors have not been able to prove this association, but they have found a relationship between hyperinsulinism during the oral glucose challenge test (OGTT) and the subsequent onset of gestational hypertension, with normal C-peptide levels. They propose a relationship between hypertension and insulin resistance and not with insulin production [37]. As described above, there is still room for further research in this field.

The association between metabolic syndrome and preeclampsia is established by an abnormal placental accumulation of glycogen, hyperinsulinemia, and subsequent insulin resistance which lead to an impairment of placental insulin signaling. We have previously demonstrated that 30% of preeclampsia patients are also diagnosed with metabolic syndrome after adapting the definition of this syndrome for pregnancy [50]. In vitro studies in cytotrophoblast cells show that hyperglycemia-induced states down-regulate angiogenic factors (VEGF, PlGF) and stimulate apoptosis, antiangiogenesis, and inflammation (higher sFlt-1, sEng, and IL-6 levels). These effects are attenuated with the use of antidiabetic drugs such as rosiglitazone [51]. Endothelial cell proliferation induced by VEGF and insulin signaling pathways are intimately related at a molecular level (MAPK and PI3K/Akt pathways) [7,10,41,45]. Alterations in signaling involving sFlt-1, PlGF, and insulin may provide cumulative damage that leads to widespread endothelial activation and injury culminating in vascular dysfunction [41]. We could conclude that hyperinsulinim and insulin resistance act synergistically with impaired angiogenic factors, so higher C-peptide levels could reflect hyperinsulinemia in pregnancy and provide value in the use of angiogenic markers as predictors of preeclampsia.

#### 4.2.2. Relationship between Lipid Metabolism and Preeclampsia

In healthy early pregnancy, insulin stimulates lipogenesis. However, in mid and late pregnancy, insulin leads to an intensification of lipolysis of fatty deposits, leading to maternal hyperlipidemia, specifically due to hypertriglyceridemia and FFA, to provide energy for the growing fetus [28,52,53,54]. In preeclampsia, there is an increase in FFA [11,52], hypertriglyceridemia [55,56], and higher oxidized-LDL in maternal serum [57,58]. Most recent studies do not find differences in cholesterol and lipoproteins in both groups, as in ours [56,59]. In our study, we have confirmed an increase in FFA in preeclampsia, in addition to a worse maternal lipid profile in LOPE than in EOPE. Lipid/lipoprotein ratios offer a better predictor for cardiovascular risk than conventional lipid components alone; they are useful in predicting plasma atherogenicity and the future risk of atherosclerosis and cardiovascular disease. In the general population, the cut-off values considered for at risk are AIP ≥ 0.24, CRR ≥ 5.0, CRI-II ≥ 3.0, and TG/HDL ratio ≥ 3.0 [60], but their predictive role in pregnant women is not established, as some lipid values during pregnancy are physiologically elevated. Recent studies show higher atherogenic ratios in preeclampsia [59]. On the other hand, elevated atherogenic ratios in pregnant women increase the risk of suffering from preeclampsia (AIP or CRR above cut-off points increase the risk 9.3 and 5.7 times, respectively) [61]. In our study, statistical significance was not reached when comparing lipid/lipoprotein ratios in preeclampsia vs. controls. However, all the atherogenic indices studied were higher (*p* < 0.05) when comparing LOPE with EOPE and only those involving triglycerides when comparing LOPE with controls. The atherogenic ratios were lower in EOPE than in controls (although without clinical relevance, as for controls they did not reach a pathological cut-off either). This could be explained by the lower gestational age studied in the EOPE group, as HDL physiologically reaches its maximum concentration in the second trimester and decreases in the third [62]. All these findings seem consistent with a different pathogenesis in LOPE and EOPE, where the former is more involved with a pre-existing metabolic syndrome [3].

In our study, we were unable to verify discrepancies in placental lipid metabolism (FAO and FAE) neither between controls vs. preeclampsia nor between EOPE vs. LOPE. This is probably due to the heterogeneity of a small sample, in which we included twin pregnancies and different degrees of severity of preeclampsia. Nonetheless, we can highlight an increase in placental triglyceride levels by comparing EOPE vs. LOPE, as recently described by Khaire et al. [59]. This could be due to greater fetal requirements in LOPE that generate placental insufficiency, increasing lipid transport and therefore a lesser amount of triglycerides accumulated in the placenta. It could be also speculated that in EOPE, there may be some difficulty in transferring both triglycerides and FFA from the placenta to the fetus due to a deficient intravillous circulation.

The relationship between dyslipidemia and preeclampsia has been described by different mechanisms. In preeclamptic patients, placental hypoxia generates oxidative stress through decreased nitric oxide and prostacyclin levels and increased free radical oxygen species [8]. This condition oxidizes LDL, producing oxysterols at higher levels than in normal pregnancies. This process occurs in up to 11% of normal pregnancies; however, it appears earlier in preeclampsia, intrauterine fetal growth restriction, or preeclampsia associated with autoimmune diseases such as systemic lupus erythematosus or antiphospholipid syndrome, and it is more frequent in EOPE than in LOPE [13,63,64]. Oxysterols promote endothelial dysfunction by different routes: they accumulate in foam cells (CD68+ macrophages) and generate large subendothelial deposits constituting “acute atherosis” of the spiral arteries, which is histologically similar to chronic atherosclerosis [3,57,65]. Acute atherosis reduces the caliber of spiral arteries, exacerbating placental hypoperfusion, causing infarction and secondary thrombotic damage [4,13]. The activation of the liver X receptors (LXRs) is another one of the complex mechanisms that control trophoblast invasion. LXRα and LXRβ are nuclear receptors that regulate lipid metabolism gene transcription by binding to oxysterols; their levels rise as the pregnancy progresses [20]. Increased oxidized-LDL produces an up-regulation of LXRs, which enhance membrane-endoglin level transcription by inhibiting the TGF-β1 signaling pathway, promoting apoptosis and inflammation [20,66,67].

A direct relationship between sEng or sFlt-1 and hyperlipidemia has not been previously established; however, in our study, a positive correlation was demonstrated between the amount of maternal triglycerides and FFA with both antiangiogenic markers and the sFlt-1/PlGF ratio. Furthermore, increases in maternal FFA predict the elevation of these factors.

Connections have been established between the mechanisms of action of antiangiogenic markers with maternal lipids, as is the case of polyunsaturated fatty acids (PUFA). The transport of PUFA to the endothelial cell is promoted by VEGF. A failure in this transport due to a decrease in functional VEGF can contribute to vasoconstriction by decreasing prostacyclin and endothelial nitric oxide and elevating the procoagulant and inflammatory response with higher thromboxane A2, IL-6, and TNF-α [16]. Moreover, an accumulation of arachidonic acid was shown in preeclamptic placentas. PUFA promote mitochondrial apoptosis with a subsequent decrease in FAO in preeclampsia [32,68], as we could demonstrate in another study from our group [25], and the accumulation of placental triglycerides [3,69] as seen in EOPE in our study. On the other hand, in vitro animal model studies indicate that omega-3 fatty acid-deficient diets may be involved in epigenetic mechanisms that lead to preeclampsia through inflammation and antiangiogenesis [70]. In addition, a dysregulation of lipoprotein lipase (LPL) activity seems to also be involved in the pathogenesis of preeclampsia, although there are still contradictory studies on its implication. In preeclampsia, there is a higher serum FFA to albumin ratio and increased LPL activity, resulting in enhanced endothelial uptake of FFA, which are further esterified and accumulated into triglycerides pools, secondary to a decrease in intracellular lipase [58,71]. Studies on LPL gene polymorphisms show variants that reduce LPL serum activity, increasing hypertriglyceridemia and decreasing HDL levels, with a probable accumulation of chylomicrons and VLDL at the arterial level leading to a greater predisposition to preeclampsia [72] or even related to more severe disease presentations such as HELLP (hemolysis, elevated liver function and low platelets) [73]. Another recent study comparing LOPE to controls shows a lower gene expression of placental LPL, but no changes in its protein expression [74].

### 4.3. Strengths of the Study

Our study is innovative in relating maternal and placental metabolism with antiangiogenic factors, obtaining promising results of correlation of sFlt-1 and sEng with increased maternal triglycerides as well as the elevation of FFA and C-peptide with sFlt-1, sFlt-1/PlGF ratio, and sEng. These findings allow for a better knowledge of the pathophysiology of preeclampsia and aid in understanding the origin of the antiangiogenic status in this disease, in which hyperinsulinemia and metabolic syndrome seem highly relevant. Moreover, the finding that FFA and C-peptide act as predictors of elevated antiangiogenic factors may help to improve the prediction of preeclampsia.

Further validation studies including energy metabolism parameters such as maternal serum FFA or C-peptide levels are needed to test the hypothesis that the inclusion of these parameters in the predictive models may help to better predict and prevent this condition.

### 4.4. Limitations of the Study

The measurement of C-peptide is difficult to interpret, and many clinical studies recommend its measurement after glucagon stimulation rather than in the fasting state. C-peptide as a biomarker offers better qualitative than quantitative information on endogenous insulin production. In addition, its excretion is purely renal and may be limited in patients with severe preeclampsia with impaired renal function.

The sample size is small and heterogeneous if we consider the inclusion of singleton and twin pregnancies, different gestational ages, and degrees of severity of preeclampsia, but it may serve as a basis for future studies to identify other predictive factors involved in the genesis of preeclampsia.

Preeclampsia is two to four times more prevalent in twin pregnancies than in single pregnancies, with earlier and more severe symptoms being more frequent in twins. The reasons for the increased relative risk of preeclampsia in twins remain unknown. The hypotheses include a higher immunologic response, relative placental hypoperfusion due to the increased placental mass, or higher production of the antiangiogenic factors simply due to the greater amount of trophoblast tissue in twin pregnancies [75,76].

The analysis on placental lipid metabolism does not confirm the decrease in FAO in preeclampsia like in previous studies [59,77,78], even in those carried out with the same methodology [25]. However, this may also be due to a heterogeneous study population.

## 5. Conclusions

Our findings of the relationship between maternal and placental energy metabolism and angiogenic markers are promising. Elevation of parameters such as FFA or C-peptide are associated to and predict the increase in some antiangiogenic markers such as sFlt-1 and sEng, and SFlt-1/PlGF ratio, respectively. For future research, the inclusion of these parameters in risk score models in different preeclampsia presentations could help us improve the predictive capacity in this condition.

## Figures and Tables

**Figure 1 nutrients-14-02172-f001:**
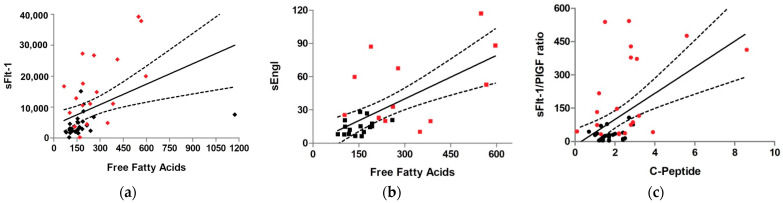
Linear regression with 95% confidence interval for prediction of angiogenic factor elevation (*p* < 0.05): (**a**) Elevation of free fatty acids (FFA) was a predictor for elevation of soluble fms-like tyrosine kinase-1(sFlt-1) (**b**) and soluble endoglin(sEng) (**c**) C-peptide was a predictor for high sFlt1/PlGF ratio. The control group is represented in black; the preeclampsia group is shown in red. The units of measurement for these parameters were: FFA (mg/dL), sFlt-1 (pg/mL), sEng (ng/mL), C-peptide (ng/mL). PIGF: placental growth factor.

**Table 1 nutrients-14-02172-t001:** Maternal, obstetric and perinatal results.

	Control *n* = 30	Preeclampsia *n* = 26	*p* Value
**MATERNAL RESULTS**			
Maternal age (years)	36.1 ± 4.3	34.5 ± 5.1	0.219
Ethnic Group:			0.592
Caucasian	18 (60.0%)	14 (58.3%)	
Hispanic	6 (20.0%)	8 (33.3%)	
Asian	0	1 (4.1%)	
Pregravid body mass index (kg/m^2^)	23.1 ± 2.7	23.8 ± 3.7	0.516
Body mass index classification:			
Underweight (<18.4)	0	1 (3.8%)	0.276
Normal weight (18.5–24.9)	19 (63.3%)	12 (46.1%)	0.052
Overweight (25.0–29.9)	3 (10.0%)	6 (23.0%)	0.166
Obesity (>30.0)	0	1 (3.8%)	0.276
Gestational weight gain (kg)	13.5 (9.00)	13.0 (6.00)	0.844
**OBSTETRIC RESULTS**			
Gestational age at study (weeks)	39.0 (2.0)	35.0 (5.0)	5.4 × 10^−8^ *
Parity:			
Primigravid	6 (22.2%)	13 (52.0%)	0.032 *
Multiparous	4 (14.8%)	5 (20.0%)	0.546
Previous C-section	14 (51.8%)	5 (20.0%)	0.068
Previous miscarriage	10 (37.0%)	7 (28.0%)	0.203
Mode of pregnancy:			0.578
Spontaneous	19 (63.3%)	18 (69.2%)	
ART (IUI)	1 (10%)	0	
ART (IVF)	5 (16.6%)	5 (19.2%)	
Twin pregnancy	4 (13.3%)	6 (23.1%)	0.342
Obstetric reason for caesarean section:			
A.-Fetal:			
Breech or transverse fetal position	10 (35.7%)	2 (8.0%)	0.016 *
Suspicion of fetal macrosomia	1 (3.5%)	0	
Intrapartum fetal distress	0	2 (8.0%)	
B.-Maternal:			
Preeclampsia	0	17 (68.0%)	1 × 10^−6^ *
Iterative C-section or Previous C-section + Bishop test ≤6	13 (46.4%)	0	8.8 × 10^−5^ *
Twin pregnancy + Bishop test ≤ 6	2 (7.1%)	0	
Labor dystocia	0	3 (12.0%)	
Elective	1 (3.5%)	0	
Fracture of femur head	1 (3.5%)	0	
Myopia magna	0	1 (4.0%)	
**PERINATAL RESULTS**			
EFW (g)	3081.4 ± 578.9	1794.1 ± 712.0	7.007 × 10^−9^ *
Centile EFW	55.5 ± 32.4	34.4 ± 37.9	0.040 *
Fetal growth restriction:			
SGA	1 (4.2%)	3 (10.7%)	0.299
IUGR	0	11 (39.3%)	2.16 × 10^−4^ *
Neonatal birth weight (g)	3232.1 ± 461.5	2092.6 ± 829.4	1.036 × 10^−7^ *
Neonatal birth centile weight	58.5 ± 26.7	37.9 ± 39.0	0.023 *
Neonatal sex:			0.868
Male	12 (37.5%)	11 (35.4%)	
Female	20 (62.5%)	20 (64.5%)	
Umbilical artery pH at birth	7.30 ± 0.06	7.28 ± 0.07	0.176
Placental weight	581.68 ± 130.85	403.37 ± 150.73	4.7 × 10^−5^ *

*Key:* C-section: caesarean section, ART: assisted reproductive technology, IUI: intrauterine insemination, IVF: in vitro fertilization, EFW: estimated fetal weight, SGA: small for gestational age, IUGR: intrauterine growth restriction. The results of qualitative variables are represented as absolute values and percentages, n (%). The results of quantitative variables are expressed as mean ± standard deviation or as the median and interquartile range (in brackets) according to the distribution of the variable. * The difference was significant compared to both groups (*p* < 0.05).

**Table 2 nutrients-14-02172-t002:** Analysis of maternal serum parameters.

	Comparison Control vs. Preeclampsia			Comparison Subsets of Preeclampsia		
	Control *n* = 30	Preeclampsia *n* = 26	*p* Value	EOPE *n* = 11	LOPE *n* = 15	*p* Value
TC (mg/dL)	257.0 ± 45.5	264.6 ± 41.0	0.532	264.4 ± 46.2	264.7 ± 38.7	0.986
LDL (mg/dL)	149.6 ± 45.5	146.6 ± 55.3	0.835	117.4 ± 63.9	165.4 ± 41.3	0.040 *
HDL (mg/dL)	71.7 ± 16.2	76.3 ± 26.1	0.463	95.2 ± 24.1	64.2 ± 19.7	0.003 *
Triglycerides (mg/dL)	257.1 ± 118.0	295.4 ± 117.9	0.258	219.1 ± 64.5	344.4 ± 119.7	0.009 *
FFA (mg/dL)	157.0 (75.0)	225.5 (202.0)	0.005 *	242.1 (226.0)	215.0 (157.0)	0.663
Cardiac risk ratio †	3.383 (2.2)	3.551 (1.4)	0.884	2.6 (1.1)	4.5 (2.2)	0.007 *
Castelli risk index II †	2.2 ± 0.9	2.2 ± 1.9	0.899	1.4 ± 0.8	2.8 ± 1.1	0.003 *
TG/HDL ratio	3.0 (2.7)	3.4 (5.2)	0.540	2.0 (1.4)	5.4 (5.7)	0.002 *
AIP †	0.17 ± 0.21	0.22 ± 0.28	0.444	0.001 ± 0.2	0.4 ± 0.2	0.001 *
FBG (mg/dL)	74.3 ± 4.9	74.1 ± 15.6	0.970	80.2 ± 17.8	69.8 ± 12.7	0.107
FINS (pmol/L)	10.5 (6.0)	13.0 (15.0)	0.197	11.0 (22.0)	14.0 (15.0)	0.918
C-peptide (ng(mL)	1.7 (0.8)	2.4 (1.4)	0.021 *	2.0 (1.3)	2.5 (1.3)	0.973
HOMA-IR †	2.0 (1.3)	2.3 (3.0)	0.393	2.4 (2.6)	2.3 (2.9)	0.785
sFlt-1 (pg/mL)	4045.5 (3850.0)	16,223.0 (19,549.0)	4 × 10^−5^ *	20,911.0 (18,504.0)	11,060.0 (15,539.0)	0.123
PlGF (pg/mL)	244.5 (352.4)	87.4 (84.1)	1 × 10^−4^ *	50.3 (54.3)	116.1 (102.5)	0.041 *
sFlt-1 /PlGF ratio	22.9 (30.2)	140.0 (363.3)	1.46 × 10^−7^ *	428.5 (644.1)	86.7 (174.9)	0.008
sEng (ng/mL)	11.6 (10.5)	42.9 (45.6)	2 × 10^−4^ *	56.7 (71.2)	32.8 (47.5)	0.827

*Key:* EOPE: early-onset preeclampsia, LOPE: late-onset preeclampsia, TC: total cholesterol, LDL: low-density lipoprotein cholesterol, HDL: high-density lipoprotein cholesterol, FFA: free fatty acids, AIP: Atherogenic Index of Plasma, FBS: fasting blood glucose, FINS: fasting insulin, HOMA-IR: homeostatic model assessment for insulin resistance, sFlt-1: soluble fms-like tyrosine kinase-1, PlGF: placental growth factor, sEng: soluble endoglin. * The difference was significant compared to both groups (*p* < 0.05). The results are expressed as the mean ± standard deviation, or as the median and interquartile range (in brackets) according to the distribution of the variable. † Cardiac risk ratio = TC/HDL; Castelli risk index II = LDL/HDL; Atherogenic Index of Plasma = Log10 (TG/HDL). HOMA-IR = [fasting serum glucose (mg/dL) × fasting insulin (mUI/L)]/405.

**Table 3 nutrients-14-02172-t003:** Analysis of lipid metabolism in placental explants.

	Comparison Control vs. Preeclampsia			Comparison Subsets of Preeclampsia		
	Control *n* = 30	Preeclampsia *n* = 26	*p* Value	EOPE *n* = 11	LOPE *n* = 15	*p* Value
Fatty Acid Oxidation (nmol/g)	3.827 (1.100)	4.374 (1.300)	0.322	4.373 (1.630)	4.409 (1.02)	0.758
Fatty Acid Esterification (nmol/g)	1.084 ± 1.124	1.370 ± 0.703	0.305	1.601 ± 0.755	1.157 ± 0.605	0.134
Triglyceride Content (mg/dL)	266.249 ± 255.726	245.660 ± 204.248	0.762	342.532 ± 217.175	156.860 ± 150.236	0.026 *
Triglyceride/Protein Ratio	762.279 ± 746.581	693.887 ± 673.828	0.744	984.714 ± 768.637	427.296 ± 457.036	0.045 *

*Key:* EOPE: early-onset preeclampsia, LOPE: late-onset preeclampsia. All metabolic processes represented are a global median between maternal and fetal sides. * The difference was significant compared to both groups (*p* < 0.05). The results are expressed as the mean ± standard deviation or as the median and interquartile range (in brackets) according to the distribution of the variable.

**Table 4 nutrients-14-02172-t004:** Correlation of maternal and placental metabolism with angiogenic factors.

	Angiogenic Factors in Maternal Plasma							
	sFlt-1		PlGF		sFlt-1/PlGF Ratio		sEngl	
METABOLISM IN MATERNAL PLASMA	rs	*p*	rs	*p*	rs	*p*	rs	*p*
Cholesterol	0.112	0.438	−0.142	0.325	0.126	0.383	0.082	0.651
LDL-c	−0.040	0.789	0.028	0.849	−0.066	0.654	-0.182	0.318
HDL-c	0.002	0.988	−0.193	0.189	0.074	0.619	0.076	0.680
Triglycerides	0.383	0.007 *	0.092	0.533	0.191	0.192	0.405	0.022 *
FFA	0.575	4.5 × 10^−5^ *	−0.080	0.606	0.509	4.13 × 10^−4^ *	0.522	0.004 *
FBS	−0.092	0.525	−0.179	0.214	0.021	0.883	-0.259	0.153
FINS	0.077	0.605	−0.207	0.162	0.167	0.263	0.215	0.245
C-peptide	0.348	0.017 *	−0.233	0.115	0.394	0.006 *	0.466	0.008 *
HOMA-IR †	0.026	0.860	−0.218	0.141	0.121	0.416	0.172	0.355
**PLACENTAL** **LIPID METABOLISM**								
Fatty acid oxidation	0.168	0.271	−0.187	0.218	0.219	0.148	0.166	0.391
Fatty acid esterification	0.193	0.203	−0.058	0.707	0.092	0.549	v0.095	0.626
Triglyceride content	−0.014	0.925	−0.071	0.645	−0.018	0.909	−0.256	0.180
TG/Protein ratio	−0.027	0.860	−0.089	0.561	−0.019	0.902	−0.257	0.178

*Key:* rs: Spearman’s correlation coefficient, sFlt-1: soluble fms-like tyrosine kinase-1, PlGF: placental growth factor, sEng: soluble endoglin, LDL-c: low-density lipoprotein cholesterol, HDL-c: high-density lipoprotein cholesterol, FFA: free fatty acids, FBS: fasting blood glucose, FINS: fasting insulin, HOMA-IR: homeostatic model assessment for insulin resistance. * The difference was significant compared to both groups (*p* < 0.05). The results are expressed as the mean ± standard deviation, or as the median and interquartile range (in brackets) according to the distribution of the variable. † HOMA-IR = [fasting serum glucose (mg/dL) × fasting insulin (mUI/L)]/405.

## Data Availability

Not applicable.

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
