# Peer review of "The Relationship between Angiogenic Factors and Energy Metabolism in Preeclampsia"

_nutrients, 2022, doi:10.3390/nu14102172_

Round 1

Reviewer 1 Report

Overall, this is a well written and very interesting manuscript. I have made some minor comments/suggestions.

Introduction

This is comprehensive and succinct.

It may be sensible to define pre-eclampsia diagnosis here in brief. The authors should recognise that fetal growth restriction is associated with placental insufficiency and a fall in PLGF.

The authors primary aim was to compare the relationship of serum antiangiogenic factors with energy metabolism parameters in maternal plasma and placenta between pre-eclamptic pregnancies and controls.

Methods

Can the authors please provide their definitions of early and late onset pre-eclampsia. As well as the severity.

My queries for the authors to please clarify:

Why did they recruit immediately prior to surgery rather than at diagnosis – a triage test at diagnosis?

Were these patients already on treatment? Can treatment affect the levels of angiogenic factors by their effect on vascular endothelium?

Table 1

  • Specific BMI definitions please
  • The controls were not gestation matched – angiogenic factors change with gestation. Please explain why?
  • The pre-eclamptic group had more fetal growth restriction – placental growth factors fall with changes in fetal growth. What diagnostic tests were used to confirm pre-eclampsia? Was the indication for birth in this group for fetal growth restriction or for pre-eclampsia?
  • Twin pregnancy is also associated with an increased risk of pre-eclampsia – this should be discussed
  • The difference in placental weight maybe due to the difference in gestation
  • What other maternal factors were present (diabetes T1/T2, antiphospholid syndrome, SLE, etc)
  • The authors state that they excluded labouring women, but the table contains the indications of caesarean section for fetal distress and labour dystocia, both of which would be diagnosed with labour.

Results

The subgroup analysis of early and late onset pre-eclampsia is very interesting. Can the authors confirm if the demographics of the subgroups was comparable (perhaps in the supplementary)?

With the difference in c-peptide, were there any diabetic women in the cohorts?

Good cardiovascular function may protect against both pre-eclampsia and fetal growth restriction (Foo FL, Mahendru AA, Masini G, Fraser A, Cacciatore S, MacIntyre DA, et al. Association Between Prepregnancy Cardiovascular Function and Subsequent Preeclampsia or Fetal Growth Restriction. Hypertension. 2018)

Patel N, Taveira TH, Choudhary G, Whitlatch H, Wu WC. Fasting Serum C‐Peptide Levels Predict Cardiovascular and Overall Death in Nondiabetic Adults. Journal of the American Heart Association.

Is it possible that some of the differences seen in lipids and c-peptide are due to backgroud differences between the cohorts? Please discuss.

Regarding the differences in FFA and c-peptide seen between the pre-eclamptic and control groups, since the concentrations of both can vary with gestation, how certain are the authors that these are due to pre-eclampsia? This may affect their predictive models later in the results section and it is worth discussing this.

Figure 1: please specify the cohort in the plots – i.e., the pre-eclamptic patients. In supplementary can the authors please include the same analysis for the controls (figure and tables) to compare.

Discussion

Overall, the discussion is very well written and structured.

My comments that the authors may want to consider are:

The argument that the predictive value of angiogenic factors in late onset pre-eclampsia can be improved by lipid and CHO metabolite levels is difficult since these were not different between late and early onset. Comparing the performance of each in LOPE and EOPE would be helpful, but the numbers are too small. Independently there was little correlation between many of the maternal and placental metabolic with angiogenic factors. This suggests that as a predictive tool at first presentation or to aid clinicians with prognosis (where PLGF and sFlt as currently used), the use of lipid and CHO metabolites is probably not helpful. However, there is potential here.

Regarding the usefulness of measuring maternal serum lipid and CHO metabolites to help clinicians make a diagnosis where it is not certain, this is also difficult to argue since in their study most of the women with pre-eclampsia were sampled prior to birth and were likely to have had birth expedited due to worsening illness (many were iatrogenic preterm births).

My impression of the data is that:

  • Altered lipid and CHO metabolism is associated with insulin resistance and inflammation in pregnancy
  • Cardiovascular health is also associated with altered lipid and CHO metabolism
  • Both are associated with an increased risk of pregnancy pathology including smaller babies and pre-eclampsia.
  • Thus, women with an altered lipid and/or CHO metabolism and poor cardio-vascular health are more likely to develop pre-eclampsia rather than being a cause of pre-eclampsia.
  • Perhaps this could guide interpregnancy/preconception interventions to reduce risk

Reviewer 2 Report

In this study, the authors analysed the relationship between antiangiogenic factors and lipid and carbohydrate metabolism in maternal plasma and placenta.

The topic is interesting and this paper provided some useful information regarding the pathogenesis of preeclampsia. The authors found the relationship between maternal and placental energy metabolism and angiogenic markers in preeclampsia. They also propose the inclusion of these parameters and FFA and C-peptide in risk score models for preeclampsia.

There are some limitations:

Perhaps the discussion is too long. The English needs minor improvement. Here are some examples:

Line 98: to those who (change to those that).

Line 105: comer after placenta.

In Material and Methods: 

No information is provided for the ELISA methods for between and within % CVs and limit of detection for the tests.

It would be interesting to examine in the paper any existing models for the calculation of the risk score for preeclampsia.

It is possible that the results of the study could be useful for some groups of specialists.

I recommend that the article be published with the revisions mentioned above. 
